# Molecular Organisation of Tick-Borne Encephalitis Virus

**DOI:** 10.3390/v14040792

**Published:** 2022-04-11

**Authors:** Lauri I. A. Pulkkinen, Sarah V. Barrass, Aušra Domanska, Anna K. Överby, Maria Anastasina, Sarah J. Butcher

**Affiliations:** 1Faculty of Biological and Environmental Sciences, Molecular and Integrative Bioscience Research Programme, University of Helsinki, 00014 Helsinki, Finland; lauri.ia.pulkkinen@helsinki.fi (L.I.A.P.); sarah.barrass@helsinki.fi (S.V.B.); ausra.domanska@helsinki.fi (A.D.); 2Helsinki Institute of Life Sciences-Institute of Biotechnology, University of Helsinki, 00014 Helsinki, Finland; 3Department of Clinical Microbiology, Faculty of Medicine, Umeå University, 90185 Umeå, Sweden; anna.overby@umu.se; 4The Laboratory for Molecular Infection Medicine Sweden (MIMS), Umeå University, 90185 Umeå, Sweden

**Keywords:** tick-borne encephalitis virus, cryo-electron microscopy, TBEV, envelope protein, membrane protein, lipid factor, glycoprotein, quasi-equivalence

## Abstract

Tick-borne encephalitis virus (TBEV) is a pathogenic, enveloped, positive-stranded RNA virus in the family *Flaviviridae*. Structural studies of flavivirus virions have primarily focused on mosquito-borne species, with only one cryo-electron microscopy (cryo-EM) structure of a tick-borne species published. Here, we present a 3.3 Å cryo-EM structure of the TBEV virion of the Kuutsalo-14 isolate, confirming the overall organisation of the virus. We observe conformational switching of the peripheral and transmembrane helices of M protein, which can explain the quasi-equivalent packing of the viral proteins and highlights their importance in stabilising membrane protein arrangement in the virion. The residues responsible for M protein interactions are highly conserved in TBEV but not in the structurally studied Hypr strain, nor in mosquito-borne flaviviruses. These interactions may compensate for the lower number of hydrogen bonds between E proteins in TBEV compared to the mosquito-borne flaviviruses. The structure reveals two lipids bound in the E protein which are important for virus assembly. The lipid pockets are comparable to those recently described in mosquito-borne Zika, Spondweni, Dengue, and Usutu viruses. Our results thus advance the understanding of tick-borne flavivirus architecture and virion-stabilising interactions.

## 1. Introduction

Tick-borne encephalitis virus (TBEV) is a member of the *Flavivirus* genus, and belongs in the family *Flaviviridae* [1]. The virus is predominantly transmitted to humans by ticks, and is the causative agent of tick-borne encephalitis, a severe disease that can lead to life-long neurological sequelae. The severity of the disease depends on the virus subtype. The European and Siberian subtypes typically cause milder symptoms than the Far-Eastern subtype that has a mortality rate of up to 40 % [2,3]. Although effective vaccines exist, the number of infections is on the rise, and there are no specific antivirals available [2,3].

The TBEV virion consists of three structural proteins, the envelope (E), membrane (M), and capsid (C) proteins, and a positive-sense single-stranded RNA genome. The genome and multiple copies of the C protein interact to form a nucleocapsid (NC). The NC is covered by the viral envelope that consists of host-derived lipids into which 180 copies of both M and E are embedded. The fundamental building block of the virion is a heterotetramer made up of two M–E dimers [4]. The heterotetramers pack tightly together to form the icosahedrally symmetric lattice of the virion surface. The E protein is the major TBEV antigen, and is responsible for receptor binding and membrane fusion [5,6,7,8,9,10]. The M protein is initially synthesised as its precursor prM, which interacts with E and protects its fusion loop from premature activation. During virus maturation, prM is proteolytically cleaved, resulting in a membrane-embedded M, and a pr peptide which dissociates from the virus [11]. The crystal structure of the E protein ectodomains and a 3.9 Å cryogenic electron microscopy (cryo-EM) reconstruction of the lab-adapted Hypr strain virion have previously revealed the organisation of the surface proteins, neutralisation epitopes and the amino acid residues probably involved in low pH-activated fusion [4,12].

Although the *Flavivirus* genus contains multiple distinct lineages, cryo-EM studies have primarily focused on the mosquito-borne flaviviruses (MBFVs), with the exception of the tick-borne TBEV and the insect-specific Binjari virus (BinJV) [4,13,14]. The BinJV structure illustrated differences between the lineages as the virion surface proteins are organised into heterohexameric spikes instead of heterotetramers like in other flaviviruses [15].

Here, we present the structure of a low-passage, European TBEV subtype (Kuutsalo-14) solved with cryo-EM and single-particle reconstruction to a resolution of 3.3 Å. The reconstruction reveals a new M protein conformation showing a potential virion-stabilising interaction between M proteins. These interactions occur across the interface between heterotetramers. M protein interactions likely compensate for the lower number of stabilising E–E contacts at the interface compared to other flavivirus structures. The residues responsible for the M–M interaction are highly conserved across TBEV isolates, suggesting that they have an important role in TBEV virion stabilisation. We clearly resolved two flaviviral lipid factors interacting with M and E peripheral and transmembrane helices. They are in similar positions to those recently described in Zika (ZIKV), Spondweni (SPONV), Dengue (DENV), and Usutu (USUV) viruses [16,17,18,19]. Furthermore, we built additional glycan residues at the E protein Asn154 glycosylation site, which has a central role in the prevention of premature fusion.

## 2. Materials and Methods

### 2.1. Cell Culture

Human neuroblastoma SK-N-SH cells (gift from Prof. Olli Vapalahti, University of Helsinki) were maintained in low-glucose Dulbecco′s Modified Eagle′s Medium (DMEM, Sigma) supplemented with 10 % foetal bovine serum (FBS, Gibco), 0.5 mg/mL penicillin, 500 U/mL streptomycin (Lonza Bioscience, Hayward, CA, USA), and 2 mM glutamine (Gibco, Waltham, MA, USA). The cells were maintained at +37 °C in a 5% CO_2_ atmosphere.

### 2.2. Virus Propagation and Titration

TBEV-Eu strain Kuutsalo-14 passage 1 (GenBank: MG589938.1, a kind gift from Prof. Vapalahti) was used to infect confluent SK-N-SH cells using a multiplicity of infection of 0.003 [20,21]. The cells were washed twice with phosphate-buffered saline (PBS), virus was added in infection medium (low-glucose DMEM, 0.5 mg/mL penicillin, 500 U/mL streptomycin, 2 mM glutamine, 2% FBS, 35 nM rapamycin [21]), and incubated at +37 °C in a 5% CO_2_. At 72 h post-infection, the supernatant was collected, and centrifuged for 10 min at +4 °C at 3800× *g*. After centrifugation, the supernatant was collected and immediately purified, or titred and stored at −80 °C.

For titration, virus samples were serially diluted 10 fold in infection medium. Confluent SK-N-SH cells in 6-well plates were washed twice with PBS, and 200 µL of the virus dilutions or infection medium was added on the cells. The cells were incubated at +37 °C for 1 h with gentle shaking every 5 min. After the incubation, 3 mL of minimum essential medium (MEM, Gibco) containing 0.5 mg/mL penicillin, 500 U/mL streptomycin, 2 mM glutamine, 2% FBS, and 1.2% Avicell was added to each well. The cells were incubated at +37 °C in a 5% CO_2_ atmosphere for 4 days, after which the medium was removed, the cells washed with PBS, and fixed with 10% formaldehyde for 30 min at room temperature (RT). The plaques were visualised by incubating the fixed cells with 0.5% crystal violet for 10 min and washing with water. The viral titre was expressed as plaque-forming units per mL (pfu/mL).

### 2.3. Virus Purification

Virus was precipitated by adding 8 % (*w*/*v*) polyethylene glycol (PEG) 8000 (Sigma-Aldrich), incubating at +4 °C with gentle mixing for 3 h, and pelleted at +4 °C, at 10,500× *g* for 50 min. The supernatant was discarded, and the pellets were carefully washed with 20 mM HEPES, 150 mM NaCl, 1 mM EDTA, pH 8.5 (HNE) and resuspended by incubating in HNE overnight at +4 °C on an orbital shaker.

The dissolved pellets were treated with 10 µg/mL RNAse A for 10 min at room temperature (RT) and loaded onto linear glycerol-potassium tartrate gradients (30% glycerol—10% glycerol, 35% potassium tartrate). The samples were centrifuged at +4 °C at 126,500× *g* for 2 h and the virus-containing light-scattering band was collected. The samples were concentrated and buffer exchanged to HNE using Amicon ultrafiltration units (MWCO 100 kDa; Millipore, Burlington, MA, USA) as per the manufacturer’s protocol at +4 °C. The titre was determined using plaque titration, and the protein content analysed using SDS-PAGE and immunoblotting. Samples were mixed with Laemmli sample buffer, incubated for 5 min at 95 °C, and proteins were resolved using electrophoresis in 4–20% polyacrylamide gels (Bio-Rad, Hercules, CA, USA). Proteins were visualised using Coomassie blue staining or by immunoblotting. Rabbit polyclonal antibodies against TBEV M were raised commercially using the peptide carboxyl-CSHAQGELTGRGHKWLEGDS-amide corresponding to the M protein residues 6 to 24 (Agrisera, Vännäs, Sweden).

For immunoblotting, the proteins were transferred onto nitrocellulose membranes (GE Healthcare, Chicago, IL, USA) using Trans Blot turbo apparatus (BioRad) using the manufacturer’s protocol. The membranes were blocked using 5% (*w*/*v*) milk in 150 mM NaCl, 20 mM Tris, 0.1% TWEEN 20, pH 7.6 (TBST) for 30 min at RT with gentle rocking and washed with TBST. Membranes were incubated with primary antibodies (1:1000 dilution in TBST containing 5% milk) at RT for 1 h with gentle rocking and washed with TBST. The primary antibodies used were monoclonal mouse anti-Langat virus E 5G5 (NR-40318, BEI Resources, [22]) and rabbit polyclonal anti-M described above. After incubation, the membranes were washed 3 times with TBST for five minutes with gentle rocking. Secondary antibodies were diluted 1:10,000 in TBST, added on the membranes, and the membranes were incubated for 30 min at RT with gentle rocking. The secondary antibodies used were IRDye 680 RD goat anti-rabbit IgG (Li-COR, Lincoln, NE, USA) and DyLight 800 goat anti-mouse IgG (KPL). After incubation, the membranes were washed with TBST and visualised using an Odyssey imager (Li-COR).

### 2.4. Inactivation and Cryo-EM Sample Preparation

Purified virus at a titre of approximately 10^10^ pfu/mL was incubated at RT for 10 min (preparation 1), with 17 uM 1-adamantylmethyl 5-aminoisoxazole-3-carboxylate (preparation 2) or with 34 uM 1-adamantylmethyl 5-amino-4-(2-phenylethyl)isoxazole-3-carboxylate (preparation 3) before inactivation to investigate the binding of these compounds to the virion [23]. Both compounds were kindly provided by Dmitry Osolodkin. The samples were inactivated by irradiation with 0.1 J/cm^2^ UV-C using a UVP Crosslinker (Analytik, Jena, Germany). The samples were vitrified on glow-discharged electron microscopy grids using a Leica EM GP plunger at 80% humidity and 1.5 s blotting time using front blotting. Lacey carbon graphene oxide-coated grids (300 micrometer grid, Electron Microscopy Sciences, Hatfield, PA, USA) were used for preparations 1 and 2, and UltrAuFoil gold grids (2/2, 300 micrometer grid, Electron Microscopy Sciences) for preparation 3.

### 2.5. Cryo-EM Data Collection

The data were collected at the SciLifeLab facilities in Umeå, Sweden (preparation 1) and Stockholm, Sweden (preparations 2 and 3) using FEI Titan Krios microscopes equipped with a Gatan K2 detector, and a Gatan K3 detector, respectively. The microscopes were operated in the counting mode. For preparation 1, movies were collected at a nominal magnification of 165,000× at a 0.82 Å^2^/pixel sampling rate with an acquisition area of 3710 by 3710 pixels and a total dose of 28.3 electrons/Å^2^ divided over 30 frames. For preparations 2 and 3, movies were collected at a nominal magnification of 81,000×, with a pixel size of 1.08 Å^2^/pixel with an acquisition area of 5760 by 4092 pixels using a total dose of 40 electrons/Å^2^ divided over 40 frames. In total, 14,135, 24,401 and 16,091 movies were collected for preparations 1, 2, and 3, respectively. See Appendix A for full data collection statistics.

### 2.6. Cryo-EM Reconstruction

Image processing was performed at the CSC—IT Center for Science Ltd. (Keilaranta, Finland) The reconstructions were performed using the Scipion 3.0 framework [24]. Each dataset was initially processed separately with a similar approach. The micrographs were aligned using the MotionCorr2 algorithm implemented in Relion 3.1 [25]. For preparations 2 and 3, movies with more than 5 pixel frame-to-frame motion, as well as movies with a total motion of more than or 15 pixel (preparation 2) or 25 pixel (preparation 3) were discarded using Xmipp movie maxshift [26,27]. Contrast transfer function (CTF) estimations were performed using CTFIND4 [28]. Micrographs with CTFs with estimated resolutions lower than 10 Å, astigmatism of more than 1000 Å, or estimated defocus outside the range of 3000–40,000 Å were discarded using Xmipp CTF consensus [24]. Particles were picked semi-automatically using Xmipp’s particle picking algorithm and extracted using Relion 3.1 with a box size of 720 (preparation 1) or 600 (preparations 2 and 3) [24,25,27]. Particles were extracted, and classified using Relion 3.1 with CTF correction, followed by auto-refinement of the best classes. [25]. Further refinement steps included per-particle CTF estimation, and Bayesian polishing [25]. The datasets for preparations 2 and 3 were merged for the final reconstruction. The local resolution was estimated using Xmipp MonoRes and Relion 3.1, and the map was locally sharpened using Xmipp LocalDeblur or minimally sharpened using the Relion 3.1 post-processing function (B-factor −5 Å^2^) [25,29,30]. Both the locally and minimally sharpened maps were used for model building.

### 2.7. Model Building

The available TBEV model (PDB: 506A) was mutated to the Kuutsalo-14 strain sequence in Coot [4,31]. The three E and M proteins that form the asymmetric unit were rigidly fitted into the locally sharpened map using UCSF ChimeraX (v. 1.3) [32,33]. The E protein ectodomain (residues 1–425) model was iteratively refined in the locally sharpened map using PHENIX version 1.20.1-4497 real space refine program, with secondary structure and non-crystallographic symmetry constraints imposed, and the ISOLDE plugin in UCSF ChimeraX (v. 1.3) [32,33,34,35]. Domain IV of the E protein (426–494) and the M protein in chains B and E that have the highest-quality density were modelled and refined as described above. The refined Domain IV region and the M protein were used as an initial model for the other chains in the asymmetric unit and fitted into the − Å^2^ B-factor sharpened map using ISOLDE [35]. The model PDBs were combined using Combine PDB files in PHENIX and the asymmetric unit refined against the locally sharpened map [34]. In the later stages of refinement, a map section containing the asymmetric unit and chains within close contact was used to refine the models. N-glycans were added using the glycan builder Linked Monomer Addition feature within Coot [31]. The restraints and coordinate files for the lipid molecules were generated using the electronic Ligand Building and Optimisation Workbench in PHENIX [34]. The lipids were rigidly fitted into the map using UCSF ChimeraX (v. 1.3) and manually refined using Coot [31,32,33]. The complete model was subjected to one iteration of real space refinement in PHENIX [34]. The geometry of the models and the map to model fit were evaluated using a combination of MolProbity and PHENIX [34,36].

### 2.8. Bioinformatic Analyses

UCSF ChimeraX (v. 1.4.dev202202182306) was used to render images of the structures [32,33]. The distances between the M–H1 helices were calculated from the centre of each helix in UCSF ChimeraX (v. 1.4.dev202202182306). Intermolecular interfaces and interacting amino acids were identified using the programs Protein Interfaces, Surfaces and Assemblies (PISA) and UCSF ChimeraX (v. 1.4.dev202202182306) [32,33,37]. Maximum distance cut-offs of 3.5 Å for hydrogen bonds and 4 Å for salt bridges were applied. The solvation free energy gain (ΔiG) was calculated using PISA [37]. The final ΔG was calculated by adding the effect of satisfied hydrogen bonds (ΔG = −0.5 kcal/mol) and salt bridges (ΔG = −0.3 kcal/mol) across the interface.

Structure-based sequence alignments were performed using PROMALS3D alignment and the RSCB PDB Pairwise Structure Alignment tool [38,39]. For the alignment between known flavivirus virion structures, only flaviviruses where both E and M were modelled to a resolution better than 4 Å were aligned (Appendix A). Sequence logos of 183 TBEV sequences from GenBank and curated by Teemu Smura, University of Helsinki (Appendix A) were generated using WebLogo3 [40].

## 3. Results

### 3.1. Cryo-EM Reconstruction of the UV-Inactivated TBEV Virion

The purified TBEV was subjected to UV-C irradiation at an energy of 0.1 J/cm, causing the virus titre to drop from 3 ± 2 × 10^10^ pfu/mL to below the level detectable by plaque assay (50 pfu/mL) (average of three repeats). Gel electrophoresis confirmed that the structural proteins were not cross-linked during this process (Appendix A).

Cryo-electron micrographs of the UV-inactivated virus showed intact spherical virions ~50 nm in diameter along with some broken particles (Appendix A). Three datasets were collected at 300 keV and reconstructed. An initial small dataset was collected on a K2 detector and reconstructed to a resolution of 3.5 Å (preparation 1). Two larger datasets were collected on a K3 detector, reconstructed independently to 3.7 Å and 3.2 Å (preparations 2 and 3, respectively). There were no significant differences between the three reconstructions. No density could be attributed to the presence of the isoxazole compounds included in the sample preparation, nor were there conformational changes in E that would accommodate the compounds. The two K3 datasets were combined, and on reclassification, did not split into two separate classes based on sample preparation, indicating their similarity. The merged K3 datasets gave a final reconstruction with a resolution of 3.3 Å (Figure 1 and Appendix A). The final reconstruction had the clearest sidechain densities and was used to build the E and M atomic models (Figure 1 and Figure 2, Appendix A). The ectodomains and the envelope leaflets were well resolved, the peripheral membrane helices and the transmembrane helices were slightly worse, but no features were distinguished in the NC (Figure 1, Figure 2 and Appendix A). Our virion reconstruction is consistent with the previously published TBEV structure of a different strain but is at a higher resolution, allowing the visualisation of side chains and their interactions [4]. We therefore conclude that UV inactivation did not cause recognisable damage to the virion structure.

The surface of the virion is composed of 30 raft-like structures, each containing three E–M–M–E heterotetramers. Each heterotetramer consists of two antiparallel E–M heterodimers (Figure 1). The asymmetric unit contains three E–M heterodimers (Figure 2). The root-mean-squared deviation (RMSD) between the current model and the published 3.9 Å model of Hypr strain is 1.6 Å for the corresponding Cα atoms [4]. The two distinguished envelope leaflets follow the polygonal shape typical of flaviviruses and the bilayer is constricted in the areas where the E and M transmembrane helices insert (Figure 1) [4,41].

The E protein is divided into four domains: three primarily β-sheet domains (domains I–III) and an α-helical domain (IV) (Figure 1). Domain II contains a hydrophobic fusion peptide at its tip (residues 100–109) [4,12]. Domain IV is membrane associated and contains three perimembrane (E–H1, E–H2 and E–H3) and two transmembrane helices (E–TM1 and E–TM2) (Figure 2). The overall folds of the E ectodomains within the virion and in the previously published X-ray crystallography model are similar, but the tip of domain II bends downwards due to the membrane curvature in the virion, leading to an RMSD of 2.1 Å across all corresponding Cα atoms [4,12]. M is proximal to domain IV of E. It consists of three α-helices: one perimembrane helix (M–H1) and two transmembrane helices (M–TM1 and M–TM2) (Figure 2).

### 3.2. Quasi-Equivalence within the Asymmetric Unit

The three E and M monomers within the asymmetric unit occupy quasi-equivalent positions (Figure 2B). Superimposition of these monomers indicates that conformational differences in the peripheral and transmembrane helices, along with the different interfaces provided by the heterotetramer ectodomain packing described previously, account for these differences [4]. The E monomers exhibit a RMSD of 1 Å, whereas the M monomers have a higher RMSD of 1.7 Å over all Cα atoms. Superimposition of the three E–M heterodimers within the asymmetric unit, based on the E protein ectodomains, indicated conformational differences between the M proteins located on the two-fold (chain D), three-fold (chain E), and five-fold (chain F) symmetry axes (Figure 2). Chain E has a Cα RMSD of 1.8 Å to chain D, and of 1.4 Å to chain F; however, chains D and F have a smaller Cα RMSD of 0.8 Å (Figure 2). The N-terminal loop regions (residues 1–24) and the C-terminal helical regions (residues 25–74) are similar between all M proteins, with low RMSD values of 0.4 Å and 0.5 Å, respectively.

The major conformational differences between the three M proteins arise due to the different locations of Asp23 and Ser24 that diverge by up to 2.8 Å at the Cα. These residues form a hinge immediately preceding the M–H1 helix, and are highly conserved in all analysed TBEV isolates (Appendix A). In comparison to chain E, the M–H1 helices are rotated anticlockwise in chains D (18°) and F (15.6°) (Figure 2). In addition, the three M–TM1 helices pack at different tilt angles relative to the E transmembrane helices. The major difference occurs at the N termini of the helices where the relative positions in chains D and F differ from chain E by 7.8 Å (Figure 2D,F).

### 3.3. Stabilising the Raft Structure—Interactions between the Heterotetramer Complexes

Previous studies on flavivirus virion structures have focused on the interactions within a single E–M–M–E heterotetramer, whereas here we describe new inter-heterotetramer interactions of the membrane regions. Chains D and F twist towards each other and interact at the heterotetramer interfaces within the raft (Figure 2B, Figure 3A and Appendix A). Each interface has a buried surface area of 173.2 Å^2^ and a free energy gain upon interface formation (ΔG) of 1.1 kcal/mol. The interaction predominantly occurs between the two M–H1 helices, which are separated by a distance of 19.1 Å (Appendix A). One π-stacking interaction between Trp37 on each M protein, and two salt bridges between Lys40 and Glu33 stabilise the M protein interface (Figure 3A).

Multiple sequence alignment of the M proteins of 183 TBEV isolates belonging to all three TBEV subtypes revealed that Trp37 and Glu33 are conserved across all analysed sequences. In addition, Lys40 is conserved in 173 of the analysed TBEV isolates. The remaining 10 isolates, including the Hypr strain, contain an arginine residue at position 40 instead (Figure 3B, Appendix A). Although arginine can also function as a hydrogen donor, no interaction between Arg40 and Glu33 is observed in the previously published Hypr strain structure [4]. Furthermore, the M proteins of Hypr do not interact at the heterotetramer interface as the M–H1 helices are separated by a larger distance of 25.7 Å (Appendix A) [4]. Analysis of all other available flavivirus structures with a resolution higher than 4 Å revealed that the M–H1 helices are separated by distances ranging from 28.9 Å to 38.6 Å (Appendix A) [4,17,18,19,42,43,44,45,46]. Consequently, none of the previous structures contain M protein interactions at the inter-heterotetramer interfaces. Structure-based sequence alignment of M proteins shows that Glu33 is conserved in insect-borne flaviviruses (Figure 3C). However, Trp37 and Lys40 are not conserved (Figure 3C, Appendix A). This suggests that the interaction can only form in tick-borne flaviviruses (TBFVs) containing the three interacting residues.

Interactions at the intra-raft heterotetramer boundary also occur between the E proteins (Figure 4, Appendix A). The total buried surface area at the E protein interface is 1095 Å^2^. The ΔG is −8 kcal/mol, of which −4.8 kcal/mol arises from hydrophobic interactions at the domain II–domain II (DII–DII) interface. At the interface, there are two π-stacking interactions between His229 of one chain and the Pro79 of the other (Figure 4A,B). In addition, we identify seven pairs of interacting residues at the domain I–domain III (DI–DIII) interfaces (Figure 4B, Appendix A). Due to the quasi-equivalence of each E protein in the asymmetric unit, DI–DIII interactions within and between asymmetric units differ (Figure 4A). We identify five interactions at the intra-asymmetric unit site: three hydrogen bonds, Tyr384-Glu170, Gln391-Ser169, Arg187-Asp380, one salt bridge Arg187-Asp380, and one π-stacking interaction (His347-Arg187) (Figure 4A). In contrast, we only identified two hydrogen bonds (Arg187-Asp380 and Gln391-Val167), and one π-stacking interaction (His347-Arg187) at the inter-asymmetric unit site. Both the E and M protein interfaces have negative ΔG values, indicating that the interactions stabilise the raft structure.

Multiple sequence alignment of the E proteins in 183 TBEV isolates shows that residues responsible for the stabilising interactions are highly conserved (Figure 4B, Appendix A). All identified interactions were also present in the Hypr structure; however, two additional hydrogen bonds and one additional salt bridge are visible in the Hypr model (Appendix A). The Hypr structure also has a higher interface surface area of 1345 Å^2^. This results in a more favourable ΔG (−14 kcal/mol) than in the Kuutsalo-14 strain.

Similar analysis of the E protein interactions was also performed for the high-resolution structures of several insect-borne flaviviruses (Appendix A). In comparison to Kuutsalo-14, all other structures had a larger total buried surface area with a higher number of E–E hydrogen bonds at the interface (Appendix A). The hydrogen bonds differed from those in TBEV consistent with sequence differences between tick-borne and insect-borne flaviviruses (Figure 4C). On average, 15 hydrogen bonds were identified in the insect-borne flavivirus structures (Appendix A). In comparison to other flavivirus structures, the TBEV Kuutsalo-14 strain forms weaker intra-raft heterotetramer interactions between the E proteins, which may increase virion flexibility and structural metastability.

### 3.4. Lipid Factors

For each E–M dimer within the asymmetric unit, the reconstruction contained two densities not accounted for by the protein model. These densities are consistent with recently described flaviviral lipid factors [16,17,18,19]. The density quality did not allow us to unambiguously identify the lipid species or fully model the fatty acid tails. As phosphatidylcholine (PC) is the most common lipid species found in flavivirus virions, we built PC headgroups and the first four carbon atoms of each fatty acid tail [47]. The PC are located near the lipid bilayer and closely associate with domain IV of the E protein and the M protein N-terminal loop. PC 1 lies between E–H1 and E–H2 (Figure 5A,B). PC 2 is located between E–H1, E–H2, E–H3, the C-terminus of E–TM2, and the N-terminal loop of M (Figure 5A,B). PISA analysis revealed that the pocket for PC 1 consists of 13–15 E protein residues (Appendix A). The surface area of the pocket is 313.6, 327.2, and 305.2 Å^2^ in chains A, B, and C, respectively. The importance of the lipids was assessed with the PISA Complex Formation Significance Score (CSS) on a scale from 0 to 1, where 0 indicates no role in complex formation and 1 indicates an essential role. CSS scores of 0.404, 0.404, and 0.463 for chains A, B, and C, respectively, indicate an auxiliary role. Additionally, PC binding is energetically favourable with ΔG values of −4.1, −7.3, and −4.7 kcal/mol for chains A, B, and C, respectively. The least solvent-accessible pocket residues are Thr406, Gly409, Arg412, Ala420, and Phe423 (Appendix A). Residues Thr406, Arg412, and Trp421 are within 4 Å of the PC 1 (Figure 4C). In chains A and B, Thr406 forms a hydrogen bond with the PC 1 proximal ester. Additionally, Arg412 forms hydrogen bonds with the PC 1 phosphate group and Trp421 interacts with the choline group via a cation-π interaction in all chains [48]. Furthermore, Ile410 is within 4 Å of PC 1, and the positive ΔG value of the residue indicates that it interacts with the fatty acid tail hydrophobically. Thr406, Arg412, and Trp421 are conserved in all analysed sequences and Ile410 only in TBEV (Appendix A).

Four M residues and 18–19 E residues form the PC 2 pocket (Appendix A). The interaction of PC 2 with the E protein chains A and C was determined to be essential (each CSS score is 0.590), but the interaction with chain B is auxiliary (CSS score 0.350). In contrast, the M protein interaction does not contribute to the complex assembly as the CSS scores were 0 for all M chains. Similarly, the energetics of the interface were only favourable for the E protein interaction. The ΔG values were −7.7, −7.9, and −5.9 kcal/mol for E protein chains A, B, and C, respectively, and of 2.2, 1.9, and 2.3 kcal/mol for M chains D, E, and F, respectively. The least solvent-accessible pocket residues were all in the E protein: Leu413, Thr414, Gly417, His438, Gly443, and Met490. Residues Thr414, His438, and Met490 in the E protein and Thr13 in the M protein were within 4 Å of PC 2. One hydrogen bond was detected between His438 of chain A and the phosphate group of the PC 2. Met490 and Thr13 had high ΔG values in all chains indicating hydrophobic interactions. His438 is conserved in all flaviviruses whereas Met490 and Thr13 only in TBEV (Appendix A).

Interestingly, a small interaction surface was observed between the fatty acid chains of the PC in chains A and C. Although the CSS scores of the interaction were low (0.05) they were energetically favourable with ΔG values of −0.5 and −0.3 kcal/mol for chain A and C, respectively. Our results indicate that the interactions of the lipids are important for the virion architecture.

### 3.5. Glycosylation

The TBEV virion contains a single glycosylation site located at Asn154 in domain I of the E protein. We observed glycan density in each E protein of the asymmetric unit [49]. We built two N-acetylglucosamines (GlcNac) with core fucosylation (Fuc) at the first GlcNac into the density, which is consistent with the TBEV glycome and with previous flavivirus structures (Figure 6) [19,50]. As expected, the glycans within the antiparallel E–E dimer cover the fusion loop (residues 100–109) of the neighbouring E monomer (Figure 6).

## 4. Discussion

As metastable particles, TBEV virions must be both sturdy enough to protect the genome and labile enough to undergo the structural changes required for fusion. Our model allows the visualisation of key determinants of TBEV metastability including raft stabilising protein–protein and protein–lipid interactions, and fusion-modulating glycosylation. Comparison between the mosquito-borne flaviviruses, TBEV Hypr and this structure shows that extrapolation between all flaviviruses from the limited number of structures available is not always justified. The coarse generalisations hold true, but the devil is in the details. A striking difference between our reconstruction and the previous published structures is the presence of an M–M protein interaction at the inter-heterotetramer interface. The interaction contributes 12% to the total energy gain upon raft formation, improving the raft stability, compensating for the weaker E protein interactions at this interface compared to the previous structures (Appendix A) [4,18,19,44].

The M protein interaction was not observed in the Hypr strain structure, which contains a single amino acid substitution in the M protein, Lys40Arg [4]. In our model, Lys40 forms two key salt bridges with Glu33 of the interacting M protein. The bulkier arginine residue may not be accommodated in this position disrupting the M–M interaction. Additionally, arginine may form a competing π-stacking interaction with Trp37 of the same chain, resulting in a different orientation of the tryptophan residue, preventing the Trp37-Trp37 interaction [51,52].

The formation of the M–M protein interaction at the intra-raft interface is dependent on the close proximity of the M–H1 helices in contrast to other published flavivirus structures (Appendix A). During flavivirus maturation, E and M undergo a large conformational change predominantly driven by rearrangement of their membrane-associated domains [19,53,54,55]. The M proteins likely come into close proximity during this process and form the observed interaction, stabilising the particle. Since the interaction interface is not observed in any mosquito-borne flaviviruses, and the interacting residues are strongly conserved only in TBEV, the M protein interface is likely unique to tick-borne flaviviruses.

The two lipid densities were reasonably modelled here as PC due to the abundance of this species in flaviviruses [47]. This was possible due to the improved resolution of the membrane compared to the Hypr strain structure where the lipid density was elusive. Previously, PC 1 position was modelled as a sphingomyelin in USUV and a phosphoceramide in DENV [17,18]. We consider both lipid classes unlikely, as phosphoceramides are exceedingly rare in flaviviruses, and our pronged density could not accommodate a single-tailed sphingomyelin molecule [47]. In the previous SPONV and USUV structures, PC 2 position was modelled as a phosphatidylethanolamine [18,19]. However, this species usually resides on the concave leaflet of curved membranes and is uncommon in flaviviviruses [56,57].

We observe that the lipid factors stabilise the mature virion and take part in the formation of the protein raft as previously suggested [16,17,18,19]. Specific deformation of the membrane curvature at these points (Figure 1) indicates that lipid rafts are probably important in assembly [58,59]. The lipid pocket-forming residues Arg412, Trp421, His438, and Met490 of E play critical role in particle formation as shown in mutagenesis studies in ZIKV, DENV, SPONV, and TBEV [16,17,19,60]. However, the functions of Thr406, Ile410 in E and Thr13 in M have not been elucidated. Since Thr406, and the presence of either an isoleucine or a leucine at site 410 are conserved throughout the flaviviruses, we propose that these residues are critical for the formation of the PC 1 pocket. In comparison, the PC 2–Thr13 interaction is only conserved in TBEV. Mutagenesis studies are required to confirm the importance of these residues in assembly.

Glycosylation of flavivirus virions plays an important role in viral particle assembly, secretion and pathogenesis, but the specific function in TBEV remains poorly understood [61,62,63,64,65,66,67]. The TBEV glycome has recently been characterised by mass spectrometry and contains a highly heterogeneous population of N-linked glycans at Asn154 of the E protein [50]. All glycans were reported to have a core of two GlcNac residues and a single mannose residue. Fucosylation of the second GlcNac was detected in approximately half of the glycan species [50]. Additionally, each glycan type had three to seven additional residues linked to the core mannose [50]. We observed density corresponding to the two first GlcNac residues and the Fuc at all glycosylation sites. This difference to the glycome analysis may be due to strain variation. The glycans are highly flexible, likely to be averaged out during the reconstruction and so this probably explains why no further density was interpretable. Importantly, the metastability of TBEV fusion is partially controlled by glycosylation protecting the fusion loop, as shown in here.

In conclusion, the overall molecular architecture of TBEV is controlled in the virion through protein–protein and protein–lipid interactions, and glycosylation. The role of the membrane is enhanced as a flexible scaffold that clusters the M and E heterodimers, allowing ectodomain and transmembrane interactions, and yet still promoting the metastability of the virus, allowing major conformational changes in the maturation pathway through to the fusion with the endosome in a newly infected cell.

## Figures and Tables

**Figure 1 viruses-14-00792-f001:**
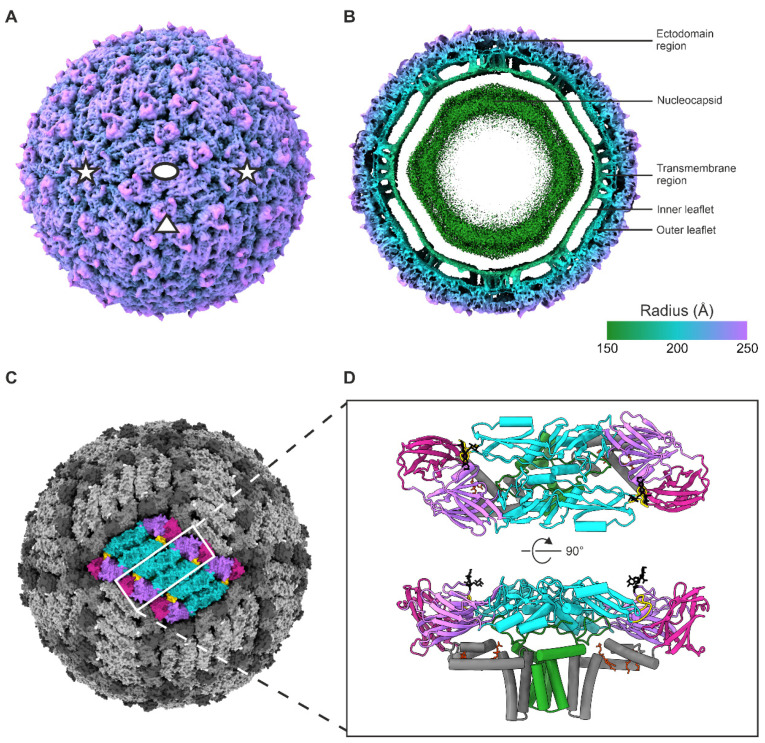
TBE virion cryoEM reconstruction and atomic model. (**A**) Radially coloured isosurface representation of the reconstruction viewed down a two-fold axis of symmetry, shown at 2.35 σ above the mean. The symmetry axes are indicated with an ellipse (two-fold), a triangle (three-fold), and star (five-fold). The colour key is the same as in panel B. (**B**) A 60 Å central slice of a radially coloured isosurface representation of the reconstruction viewed down a two-fold axis of symmetry, at 2.35 σ above the mean. (**C**) Surface representation of the reconstruction, with a raft of three E–M–M–E heterotetramers highlighted, viewed down a two-fold axis of symmetry. (**D**) Cartoon representation of the E–M–M–E heterotetramer shown from the top and side views. (**C**,**D**) The E domains are coloured in purple (domain I), turquoise (domain II), magenta (domain III), grey (domain IV), the E fusion loop is coloured in gold, the E Asn154 glycan motif in black, the M protein in green, and the phosphatidylcholine (PC) lipids in red-orange.

**Figure 2 viruses-14-00792-f002:**
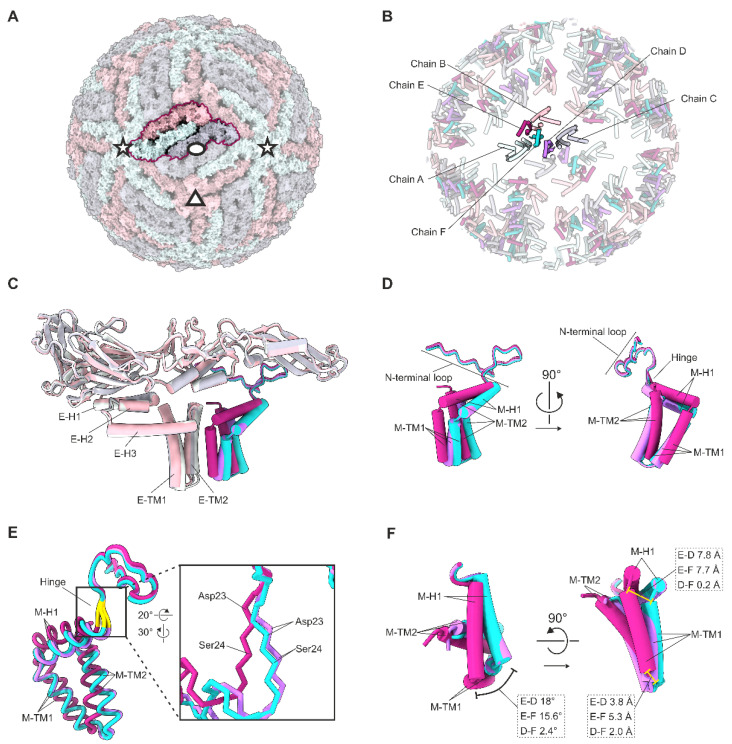
Quasi-equivalence within the asymmetric unit. (**A**) Surface representation of the TBEV virion. One asymmetric unit is highlighted. The symmetry axes are indicated with an ellipse (two-fold), a triangle (three-fold), and stars (five-fold). (**A**–**C**) The three E proteins forming the asymmetric unit are coloured in light green (chain A around the five-fold axes of symmetry), pink (chain B around the three-folds), and grey (chain C around the two-folds). (**B**) Cartoon representation of the E and M membrane-associated domains within the context of a virion. (**B**–**F**) The three M proteins within one asymmetric unit are coloured in purple (chain D), magenta (chain E), and turquoise (chain F). (**C**) Superimposition of the three E–M heterodimers that make up the asymmetric unit, based on the E proteins. Transmembrane helices of E are labelled as E–TM1, and E–TM2; perimembrane helices are labelled E–H1, E–H2, and E–H3. (**D**) Detailed view of the superimposed M proteins of one asymmetric unit. Transmembrane helices of M are labelled as M–TM1, and M–TM2; perimembrane helix is labelled M–H1; N-terminal loop and hinge are indicated. (**E**) Superimposition of the three M proteins in one asymmetric unit as in (**C**). The hinge between M–H1 and the N-terminal loop is highlighted in yellow and is shown zoomed in. The differences in the backbone positions at residues 23 and 24 are indicated. (**F**) Superimposition of the three M proteins in one asymmetric unit as in (**C**) with the N-terminal residues 1–21 hidden for clarity. The angles between the superimposed helices are shown in the top view (left), and the distances between the corresponding Cα atoms are indicated in the side view (right).

**Figure 3 viruses-14-00792-f003:**
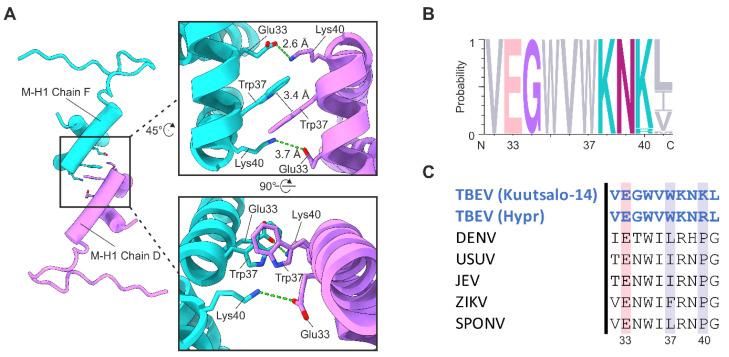
Interactions at the M heterotetramer interface. (**A**) Interaction between the two tilted M proteins located at the heterotetramer interface. Chain D is shown in purple and chain F shown in turquoise. Interacting residues are shown as stick representations and labelled in the close-up views (right). Hydrogen bonds are shown by green dotted lines. The distance between the interacting residues is labelled. (**B**) Sequence conservation of the interacting M residues in 183 TBEV isolates. The consensus residues are shown with the size of the letter proportional to the frequency of the residue. Interacting residues are marked with numbers and the residues are coloured according to chemistry (grey, hydrophobic; pink, acidic; purple, polar including glycine; turquoise, basic; magenta, neutral). (**C**) Structure-based sequence alignment of flaviviruses. Positions conserved throughout flaviviruses are highlighted in pink, and positions conserved only in the majority of TBEV strains are highlighted in grey. The interacting residues are numbered according to TBEV Kuutsalo-14 sequence (GenBank: MG589938.1).

**Figure 4 viruses-14-00792-f004:**
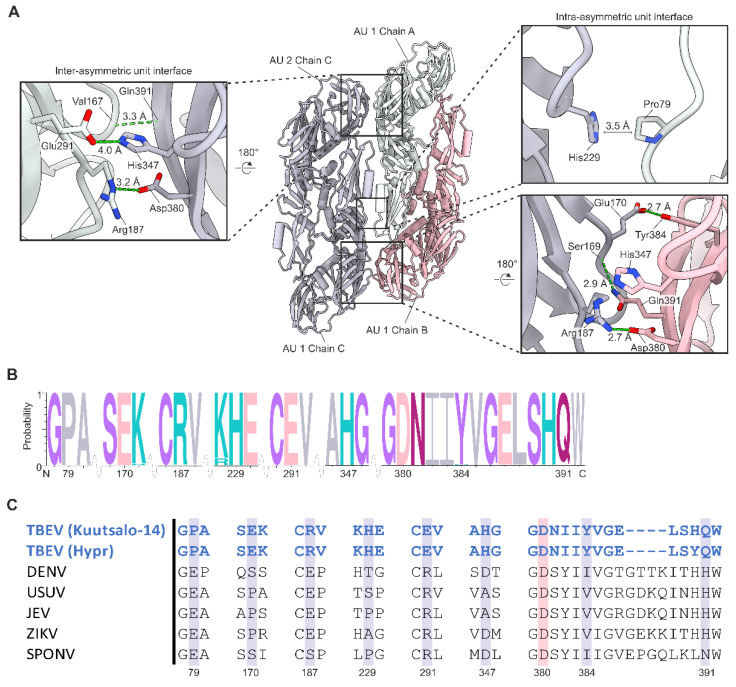
Interactions at the E heterotetramer interface. (**A**) Interactions between the E proteins located at the heterotetramer interface. Chain B is shown in pink, chain A in light green and chain C in grey. The asymmetric units (AU) are labelled. Interacting residues are shown as stick representations and labelled in the close-up views. Hydrogen bonds are shown by green dotted lines. The distance between the interacting residues is labelled. The π-stacking interaction between His347 and Arg187 and the salt bridge between Arg187 (atom-NE) and Asp380 are not shown for clarity. His347 and Arg187 are separated by a distance of 3.4 Å, and Arg187 (atom-NE) and Asp380 by 4 Å. (**B**) A consensus sequence of the interacting E residues in 183 TBEV sequences. Letter size is proportional to the frequency of the residue. Interacting residues are marked with numbers and the residues are coloured according to chemistry (grey, hydrophobic; pink, acidic; purple, polar including glycine; turquoise, basic; magenta, neutral). The wave indicates a sequence break. (**C**) Structure-based sequence alignment of flaviviruses with high-resolution structures. Residues conserved throughout flaviviruses are highlighted with pink, and residues conserved in TBEV isolates are highlighted in grey. The interacting residues are numbered according to TBEV Kuutsalo-14 sequence (GenBank: MG589938.1).

**Figure 5 viruses-14-00792-f005:**
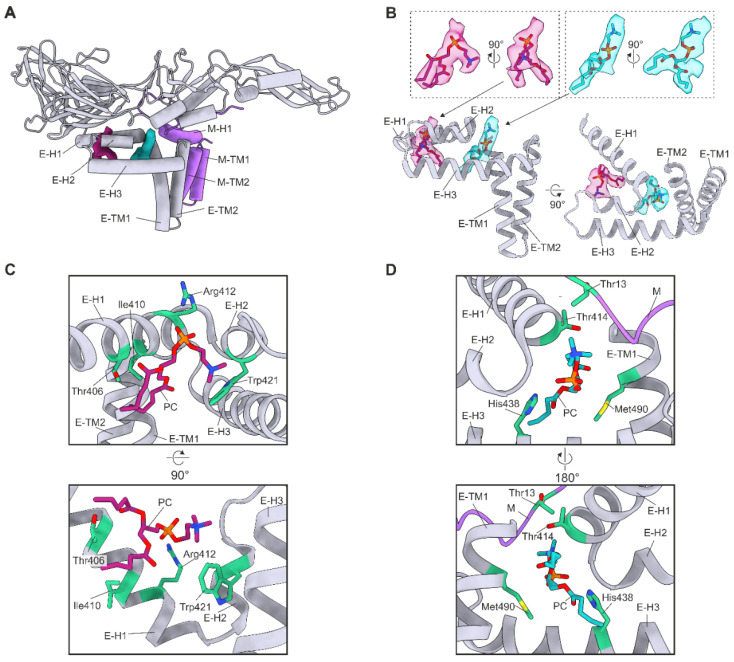
Ordered phosphatidylcholine (PC). (**A**) Two PC molecules bound per heterodimer. (**B**) Location of the lipid factors in relation to the E protein Domain IV. The PC atomic models are coloured in magenta (PC 1) and turquoise (PC 2) with oxygen, nitrogen, and phosphorus atoms coloured in blood red, blue, and orange, respectively. (**A**,**B**) The E chain C is coloured in grey, and the M chain D is coloured in purple. The densities corresponding to PC 1 and PC 2 are coloured in magenta and turquoise, respectively. The transmembrane (TM) and perimembrane (H) helices of E and M are labelled. (**C**) Residues within 4 Å of PC 1. (**D**) Residues within 4 Å of PC 2. M protein is coloured in purple. (**C**,**D**) Residues within 4 Å of PC 1 are shown, labelled, and coloured in green.

**Figure 6 viruses-14-00792-f006:**
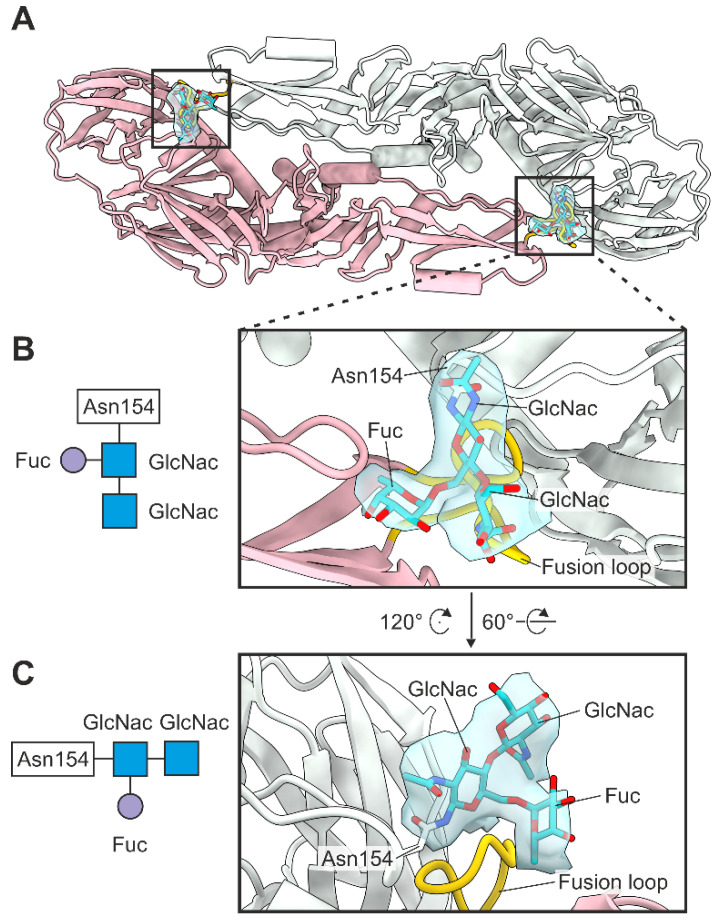
E protein glycosylation. (**A**) Top view of the E–E dimer. Chain A is coloured in light green and chain B in pink. Fusion loop (residues 100–109) is coloured in gold, and the glycan residues are coloured in turquoise with oxygens coloured in red and nitrogens in blue. The density corresponding to the glycans is shown in light blue. (**B**,**C**) Close-ups showing the position of the glycans relative to the fusion loop from two angles with Asn154 and the glycan residues labelled. The glycosylation of Asn154 is shown schematically to the left.

## Data Availability

The raw electron microscopy data has been deposited to Electron Microscopy Public Image Archive (EMPIAR). The reconstructed volumes have been deposited to the Electron Microscopy Database (EMDB) with the accession codes EMD-14516 and EMD-14512 for preparation 1 and the final reconstruction, respectively. The atomic coordinates of the model have been submitted to the Worldwide Protein Data Bank (wwPDB) with the PDB ID 7Z51.

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
