# Peer review of "Molecular Organisation of Tick-Borne Encephalitis Virus"

_viruses, 2022, doi:10.3390/v14040792_

Round 1

Reviewer 1 Report

The authors provide very accurate cryo electron microscopy structural investigation of flavivirus TBEV. The high 3.3 Å resolution allowed the authors to build atomic models of viral structural proteins M and E, including reconstruction of their transmembrane domains. The structure reveals two lipids bound in the E protein, which are important for virus assembly. Similar lipid pocket factors were previously described for several mosquito-borne alpha- and flaviviruses. The authors found these features for the Kuutsalo-14 isolate for the first time. Their results really advance the understanding of tick-borne flavivirus architecture and virion-stabilising interactions.

So far, lipids and their significance in the virus lifecycle have been underestimated. I am very pleased that the high-resolution structural studies of enveloped viruses published over the past few years have brought us very close to identifying specific roles for various lipid molecules. In addition, the new features of E protein glycosylation were found. 

The work is significant in its content and demonstrate excellent figures. My only criticism is regarding Legends to Figures 3B and 4B saying that "...the residues are coloured according to chemistry (grey, neutral; pink, acidic; purple, polar; turquoise, basic; magenta, hydrophobic). I can't agree that the residues depicted in magenta (G, S, C, Y) are hydrophobic. It is more right to name them "polar". In order to differentiate these residues from the amid group containing "N, Q" polar residues (depicted in purple) one can name them as "OH-/SH-group containing polar residues", and the glycine is attributed to this group by the WebLogo service just by default. In contrast, the residues depicted in grey (P, A, V, I, L, W) could rather be called "hydrophobic" instead of "neutral".

Author Response

Reviewer 1

“The authors provide very accurate cryo electron microscopy structural investigation of flavivirus TBEV. The high 3.3 Å resolution allowed the authors to build atomic models of viral structural proteins M and E, including reconstruction of their transmembrane domains. The structure reveals two lipids bound in the E protein, which are important for virus assembly. Similar lipid pocket factors were previously described for several mosquito-borne alpha- and flaviviruses. The authors found these features for the Kuutsalo-14 isolate for the first time. Their results really advance the understanding of tick-borne flavivirus architecture and virion-stabilising interactions.

So far, lipids and their significance in the virus lifecycle have been underestimated. I am very pleased that the high-resolution structural studies of enveloped viruses published over the past few years have brought us very close to identifying specific roles for various lipid molecules. In addition, the new features of E protein glycosylation were found.

The work is significant in its content and demonstrate excellent figures. My only criticism is regarding Legends to Figures 3B and 4B saying that "...the residues are coloured according to chemistry (grey, neutral; pink, acidic; purple, polar; turquoise, basic; magenta, hydrophobic). I can't agree that the residues depicted in magenta (G, S, C, Y) are hydrophobic. It is more right to name them "polar". In order to differentiate these residues from the amid group containing "N, Q" polar residues (depicted in purple) one can name them as "OH-/SH-group containing polar residues", and the glycine is attributed to this group by the WebLogo service just by default. In contrast, the residues depicted in grey (P, A, V, I, L, W) could rather be called "hydrophobic" instead of "neutral".”

Response:

We thank the reviewer for this observation. The figure legends for 3B and 4B had indeed been butchered and so they are now corrected. Figure legends for figure 3 B and figure 4 B have been changed to reflect the true chemistry of the side chains following the WebLogo3 nomenclature. Glycine being a part of the “polar” group has been indicated. The text at line 318 now reads: “Interacting residues are marked with numbers and the residues are coloured according to chemistry (grey, hydrophobic; pink, acidic; purple, polar including glycine; turquoise, basic; magenta, neutral).” The text at line 358 now reads: “Interacting residues are marked with numbers and the residues are coloured according to chemistry (grey, hydrophobic; pink, acidic; purple, polar including glycine; turquoise, basic; magenta, neutral).”

Reviewer 2 Report

In this work, Pulkkinen and colleagues determined the structure of a low passage isolate of tick-borne encephalitis (TBE) virus by cryo-EM to a resolution of 3.3 Å. While the structure conforms to the basic organization of mature flavivirus particles in general, the study revealed several new details, especially in comparison to a previously published lower resolution cryo-EM structure of a mouse-adapted and high passage laboratory strain of TBE virus (Hypr). Details of the new structure were analyzed meticulously in comparison to that of other flaviviruses published previously and revealed potentially virus-stabilizing interactions between M proteins that had not been described so far. Importantly, the authors were also able to resolve two interaction sites of M and E with lipids in the viral membrane, corroborating the presumed role of lipid pockets in the viral envelope proteins for stabilizing mature virions.

Overall, the work conducted as well as data analyses are of high quality and the results presented will certainly be valuable to those interested in the structural details underlying the functions of flavivirus particles.

Minor comment:

Lines 136 – 139 (Materials and Methods): Considering that sample preparation is critical for obtaining high resolution reconstructions, some comment on the rationale for incubating the purified virus with the two isoxazole compounds before UV-inactivation would be valuable for those interested in relevant technical details.

Author Response

Reviewer 2

“In this work, Pulkkinen and colleagues determined the structure of a low passage isolate of tick-borne encephalitis (TBE) virus by cryo-EM to a resolution of 3.3 Å. While the structure conforms to the basic organization of mature flavivirus particles in general, the study revealed several new details, especially in comparison to a previously published lower resolution cryo-EM structure of a mouse-adapted and high passage laboratory strain of TBE virus (Hypr). Details of the new structure were analyzed meticulously in comparison to that of other flaviviruses published previously and revealed potentially virus-stabilizing interactions between M proteins that had not been described so far. Importantly, the authors were also able to resolve two interaction sites of M and E with lipids in the viral membrane, corroborating the presumed role of lipid pockets in the viral envelope proteins for stabilizing mature virions.

Overall, the work conducted as well as data analyses are of high quality and the results presented will certainly be valuable to those interested in the structural details underlying the functions of flavivirus particles.

Minor comment:

Lines 136 – 139 (Materials and Methods): Considering that sample preparation is critical for obtaining high resolution reconstructions, some comment on the rationale for incubating the purified virus with the two isoxazole compounds before UV-inactivation would be valuable for those interested in relevant technical details.”

Response:

Part of the original aim of the work was to study the binding mode of two isoxazole compounds to TBEV virions with cryo-EM. The highest-resolved reconstructions with either compound or with the combined data set did not contain any identifiable density attributable to the compounds in question and our laboratory tests did not indicate any inhibitory activity on the virus. The compounds have previously been proposed to bind to the β-OG pocket of E when it is in an open conformation and to inhibit the viral infection. We did not observe that open conformation in any of the reconstructions. Neither compound appeared to have any effect on the resolution obtained, as a small data set collected without the compounds present, also went to a similar resolution, and displayed the same features as the final dataset. The apo dataset was not combined with the two others as it was taken under different imaging conditions, including a very different sampling, not because the resulting structure looked any different. We do not consider that the compounds had any effect on getting to higher resolution, rather this was due to small cumulative effects of careful sample preparation, improved detectors, the very large dataset, and improvements in image processing that have been made over the past few years, seen as a general trend in single particle image processing when one looks at depositions in the EMDB. We have modified the text in both the Materials and Methods (lines 136-137) and the Results (lines 224-226).

The text now reads starting at line 136:  “Purified virus at a titer of approximately 1010 pfu / ml was incubated at RT for 10 min (preparation 1), with 17 uM 1-adamantylmethyl 5-aminoisoxazole-3-carboxylate (preparation 2) or with 34 uM 1-adamantylmethyl 5-amino-4-(2-phenylethyl)isoxazole-3- carboxylate (preparation 3) before inactivation to investigate the binding of these compounds to the virion [23].

and at line 224

There were no significant differences between the three reconstructions. No density could be attributed to the presence of the isoxazole compounds included in the sample preparation, nor were there conformational changes in E that would accommodate the compounds. The two K3 datasets were combined, and on reclassification, did not split into two separate classes based on sample preparation, indicating their similarity.  The merged K3 datasets gave a final reconstruction with a resolution of 3.3 Å (Figure 1, S2 B-C).”